# Bioreactivity, Guttation and Agents Influencing Surface Tension of Water Emitted by Actively Growing Indoor Mould Isolates

**DOI:** 10.3390/microorganisms8121940

**Published:** 2020-12-07

**Authors:** Maria A. Andersson, Johanna Salo, Orsolya Kedves, László Kredics, Irina Druzhinina, Jarek Kurnitski, Heidi Salonen

**Affiliations:** 1Department of Civil Engineering, Aalto University, Box 12100, FI-00076 Aalto, Finland; johanna.72salo@gmail.com (J.S.); jarek.kurnitski@aalto.fi (J.K.); heidi.salonen@aalto.fi (H.S.); 2Department of Microbiology, Faculty of Science and Informatics, University of Szeged, Közép Fasor 52, 6726 Szeged, Hungary; varga_orsi91@yahoo.com (O.K.); kredics@bio.u-szeged.hu (L.K.); 3The Key Laboratory of Plant Immunity, Jiangsu Provincial Key Lab of Solid Organic Waste Utilization, Nanjing Agricultural University, Nanjing 210095, China; irina.druzhinina@njau.edu.cn; 4Fungal Genomics Laboratory (FungiG), Nanjing Agricultural University, Nanjing 210095, China; 5Microbiology Group, Research Area Biochemical Technology, Institute of Chemical, Environmental and Bioscience Engineering (ICEBE), TU Wien, A1060 Vienna, Austria; 6Department of Civil Engineering and Architecture, Tallinn University of Technology, Ehitajate Tee 5, 19086 Tallinn, Estonia

**Keywords:** bioreactivity, fungi, guttation, indoor mould

## Abstract

The secretion of metabolites in guttation droplets by indoor moulds is not well documented. This study demonstrates the guttation of metabolites by actively growing common indoor moulds. Old and fresh biomasses of indoor isolates of *Aspergillus versicolor*, *Chaetomium globosum*, *Penicillium expansum*, *Trichoderma atroviride*, *T. trixiae*, *Rhizopus* sp. and *Stachybotrys* sp. were compared. Metabolic activity indicated by viability staining and guttation of liquid droplets detected in young (<3 weeks old) biomass were absent in old (>6 months old) cultures consisting of dehydrated hyphae and dormant conidia. Fresh (<3 weeks old) biomasses were toxic more than 10 times towards mammalian cell lines (PK-15 and MNA) compared to the old dormant, dry biomasses, when calculated per biomass wet weight and per conidial particle. Surfactant activity was emitted in exudates from fresh biomass of *T. atroviride*, *Rhizopus* sp. and *Stachybotrys* sp. Surfactant activity was also provoked by fresh conidia from *T. atroviride* and *Stachybotrys* sp. strains. Water repealing substances were emitted by cultures of *P. expansum*, *T. atroviride* and *C. globosum* strains. The metabolic state of the indoor fungal growth may influence emission of liquid soluble bioreactive metabolites into the indoor air.

## 1. Introduction

Fungal hyphae sense mechanical and optical stimuli and digest before they ingest. Metabolically active growing fungal hyphae are constantly secreting degrading enzymes and absorbing nutrients. Fungal hyphae also communicate with other microorganisms in their surrounding by secreting secondary metabolites as signaling molecules [1,2]. These small effector molecules, including mycotoxins, antimicrobials, surfactants and small secreted cysteine-rich proteins such as hydrophobins or cerato-platanins, may regulate fungal development as hyphal growth, sporulation, germination of conidia, biofilm formation, and virulence, including parasitism on other fungi [3,4,5,6,7,8,9,10,11,12].

Respiratory illness can be associated with exposure to moulds that develop in dampness [13,14,15,16]. Fungal proteins and enzymes may induce IgE-mediated asthma, but fungi may also act as potent immune modulators promoting asthma independent of antigenic activity [17]. Fungal secondary metabolites and β-glucans may alter the innate immunity of humans regulating inflammasome formation and cytokine release [18,19,20,21,22,23]. Surfactants in indoor air and dust may cause irritation and instability of mucous membranes, and may strengthen the effects of bioreactive compounds, which may result in IgE-mediated allergy or asthma [24,25,26]. However, a consensus concerning the exact causative mechanism or agents behind the respiratory diseases recorded in moist buildings has not yet been achieved [27,28,29,30,31]. No guidelines for unhealthy levels of indoor mould exposure have been defined [32,33,34].

The specific “mould odour” associated with the excretion of volatile organic compounds (VOCs) by indoor fungi [34,35] has been connected to unhealthy indoor air and considered as a possible indicator of active fungal growth [16]. Fungal hyphae may also guttate liquid droplets exudating secondary metabolites or waste products [8,9,36,37,38]. Compared to conidia, hyphal fragments and fungal DNA, liquid guttation droplets and their connection to metabolic activity of indoor fungi have gained little attention during the monitoring of indoor air quality. No information is available about surfactants in guttation droplets of indoor fungi, their effects on the dispersal of other airborne indoor contaminants, and their effect on the water availability for fungi growing on indoor building materials. Metabolic activity in fungal biomass in liquid cultures of food spoiling fungi has been monitored by a voltametric electronic tongue (smart tongue) [39] but the impact of metabolic activity of indoor fungi on indoor air quality is not yet understood.

The aim of our study was (a) to develop and evaluate fast microscopic and toxicological methods for monitoring metabolic activity in fungal colonies, and (b) to reveal the metabolic difference between fresh, actively growing and old, desiccated, dormant colonies growing on building materials and culture media.

## 2. Materials and Methods

### 2.1. Experimental Design

Strains representing common indoor mould species and genera were selected for the development and evaluation of methods for monitoring metabolic activity. A scheme illustrating the experimental design is shown in Figure 1.

### 2.2. Identification of the Fungal Strains 

Settled dust, indoor building materials and inlet air filters were collected from urban buildings connected to indoor air complaints. Characterization of the indoor microbial isolates proceeded as follows: biomass lysates of fungal colonies were initially categorized based on toxic responses in two toxicity assays: BSMI (boar sperm motility inhibition assay) and ICP (inhibition of cell proliferation assay) and autofluorescence as described earlier [38,40]. The obtained categories were further identified as morphotypes down to the genus level based on colony morphology on MEA, ability to grow at 37 °C, conidiophore morphology and conidial size as seen under light microscope (Olympus CKX41, Tokyo, Japan; magnification 400×, image recording software Cellsense^®^ standard version 11.0.06) [40,41,42]. Selected representatives of the morphotypes were identified during previous studies by ITS and *tef1α* sequencing with the primer pairs ITS1 (5′-TCCGTAGGTGAACCTGCGG-3′)/ITS4 (5′-TCCTCCGCTTATTGATATGC-3′) and EF595F (5′-CGTGACTTCATCAAGAAGATG-3′)/EF1160R (5′-CCGATCTTGTAGACGTCCTG-3′), respectively (see Table 1 for GenBank accession numbers of the resulted sequences), followed by sequence analysis performed with Nucleotide BLAST (https://blast.ncbi.nlm.nih.gov), and/or identified by DSMZ (Deutsche Sammlung von Mikroorganismen und Zellkulturen, Braunschweig, Germany) or WI (Westerdijk Institute, Wageningen, The Netherlands).

### 2.3. Cultivation of Mould Colonies

The methods used for cultivating the mould colonies on 35 mL MEA (15 g malt extract from Sharlab, Spain, and 12 g of agar from Amresco, Solon, OH, USA, in 500 mL of H_2_O) per plate were described previously. Actively growing and dehydrated, dormant biomass was prepared as follows: the strains were cultivated on MEA for 1–3 weeks and for 6 months, respectively. The plates were sealed with gas-permeable adhesive tape to slow moisture loss during the culturing at a relative humidity (RH) of 30–40% and a temperature of 22–24 °C and stored with the lid upwards.

### 2.4. Viability Staining with Hoechst 33342 + Propidium Iodide

Old, dormant and actively growing aerial fungal hyphae were stained with the viability stain Hoechst 33342 + PI. Hoechst 33342 crosses intact and damaged plasma membranes, staining the DNA in nuclei and mitochondria blue in live and dead cells. PI crosses only plasma membranes with disturbed or exceptional permeability, staining DNA and RNA in affected cells red [37,43,44]. The staining protocol was as follows: fungal biomass, old and new, cc. 20 mg was mixed with 200 µL water containing 10 µg µL^−1^ PI and Hoechst 33342, respectively, and incubated in the dark for 10–15 min at 37 °C. The stained aerial hyphae were inspected with a fluorescence microscope (Nikon Eclipse E600; Nikon Corporation, Tokyo, Japan) at 400× magnification, with filters of BP 330 nm to 380 nm (excitation) and LP 480 nm (emission). Confirmation of the blue staining by Hoechst 33342 and red staining by PI was performed as follows: fungal biomass stained with Hoechst 33342 emitted only blue fluorescence ex BP330-380 nm/LP em 480 nm. Red fluorescence thus was provided by intracellular PI. The exhaust filter was stained dry and after having been covered by agar for 24–48 h. A piece of the dry filter and of the filter which had been moisturized by embedding in agar was soaked in 500 µL staining solution and treated as described above.

### 2.5. Toxicity Test Using Biomass Lysates

The toxicity tests measured: (a) the toxins affecting the cellular energy metabolism, the mitochondria and ion homeostasis based on the inhibition of the motility of boar spermatozoa (BSMI) [40,41,45,46,47]; and (b) the toxins affecting macromolecular synthesis and cytostatic activity based on the inhibition of the proliferation of the somatic cell line PK-15 (ICP) [40,45,46]. Briefly, the BSMI assay used commercial boar semen extended to 14% in the extender MR-A (Kubus S.A. Madrid, Spain), obtained from Figen Ltd. Tuomikylä, Finland, containing 27 × 10^6^ sperm cells mL^−1^. The sperm cells were exposed at room temperature 30–60 min, before motility was induced by incubation at 37 °C for 5 min. For the ICP assay, the cells were grown in RPMI 1640 (complete medium), exposing for 2–3 days, 2–4 × 10^4^ cells in 200 µL medium per the wells in a 96-microtiter plate.

In this study, the ICP test was also used for comparing toxicity in old, dormant, and new, actively growing fungal biomasses. The procedure of the ICP test using the primary cell line of PK-15 cells and the malignant cell line of MNA cells, was briefly as follows: 60 mg of plate grown biomass was suspended in 600 μL of ethanol, heated in a water bath to 55–60 °C for 10 min and stored at room temperature for 5 h. The lysates were diluted by two-fold, exposing the target cells growing in 96-well microtiter plates. The toxicity endpoints for the old and new fungal biomass were measured by fluorimetric readouts of resazurin reduction combined with microscopic inspection of cell viability and intactness of the cell monolayer as described earlier [46]. Fluoroskan Ascent, Thermo Scientific, Vantaa, Finland, ex544 nm/em 590 nm was used for fluorometry. An Olympus CKX41 microscope (Tokyo, Japan; magnification 400×–100×, image recording software: Cellsense^®^ standard version 11.0.06) was used for inspection of the cell monolayer. The toxicity endpoint EC_50_ was expressed as the lowest concentrations where the ratio of the living cells was less than 50%. All of the mammalian somatic cell toxicity assays were repeated three times. The EC_50_ obtained by the resazurin reduction and the microscopic examination fitted between EC_90_ and EC_10._ The maximal difference between the two methods was one dilution step.

The half maximal effective concentrations (EC_50_) for the biomass lysates were calculated as µg wet weight of fungal biomass per mL of test cell suspension. The difference between three repeated tests was within one dilution step.

### 2.6. Toxic Responses in Old Biomass of Stachybotrys sp. Grown for >20 Years on Paper Board

Toxic responses of old, dormant, and young, fresh fungal biomasses were compared by the procedure described below. A piece of paperboard (10 cm × 10 cm) sampled 1996 was stored in a sealed glass jar at room temperature until 2017. The old dry biomass scraped from the paperboard from 3 different locations were (a) cultivated on 10 MEA plates, (b) dissolved in ethanol (60 mg mL^−1^). One colony was pure-cultured and named strain HJ5.

### 2.7. Monitoring Guttation

Guttation was monitored and compared in actively growing and old, dormant fungal biomasses as follows: cultures grown for 1 to 3 weeks or 6 months grown on MEA and gypsum liner in sealed Petri-plates were stored with the lid upwards. The plates with actively growing fungi were inspected for guttation in stereomicroscope every second day during 3 weeks of growth. To facilitate visualization of guttation, the plates were also inspected under UV light when the guttation droplets often emitted autofluorescence. Cultures grown for 6 months were inspected for guttation every second day after 5.5 months of incubation.

### 2.8. Monitoring Emission of Surface-Active Compounds

Fungal cultures grown for 1–3 weeks on MEA and gypsum board were stored with the lid upwards. Droplets condensed on the inner surface of the Petri-plate lids were inspected regularly with stereomicroscope in visible and UV light. The size of the droplets and the water free zone around condensed droplets were measured by Dino-lite portable microscope with Dinocapture imaging software (Dino-Lite Europe/IDCP B.V. Almere, The Netherlands). Hydrophobic water-repelling activity was recorded as visible, water-free zone around substances emitted by the fungal culture. Surfactant activity was recorded as increased surface of droplets exposed to fungal substances. Measurement of reduction of surface tension, i.e., the increase of the surface of a drop of pure water of 20–500 µL after addition of 5–50 µL of exudates or conidial dispersals was described previously [48].

## 3. Results

### 3.1. Comparison of Methods Monitoring Metabolic Activity in Fungal Biomass

Methods for estimating differences in metabolic activity between fresh, actively growing, and old, dehydrated fungal biomasses were compared. The metabolic activity was monitored by (1) viability staining with fluorescent dyes, (2) guttation, (3) toxic response in an in vitro bioassay, and (4) secretion of surface-active substances. Isolates representing four common toxigenic indoor genera, *Trichoderma*, *Stachybotrys*, *Aspergillus* and *Chaetomium* were investigated. The strains, their origin and identification are presented in Table 1.

#### 3.1.1. Fluorescence Intensity in Actively Growing and Dormant Fungal Biomass Stained with the Viability Staining by Hoechst 33342 Combined with Propidium Iodide

Young and old fungal biomasses were stained with the live-dead stain Hoechst 33342 + propidium iodide (PI). Fungal biomass stained with Hoechst 33342 only emitted exclusively blue fluorescence. Red fluorescence thus was provided by intracellular PI. Young, actively growing hyphae and conidia of A. versicolor are compared with old ones in Figure 2. The Figure shows that the young hyphae have active organelles containing fluorescent nucleic acids and performing metabolic processes. The old fungal structures show dehydrated, degraded hyphal structures impermeable to both of the stains, therefore emitting no visible fluorescence. Phase contrast micrographs of young mycelia guttating liquid droplets and old, dehydrated biomass of *Stachybotrys* sp. are shown in Figure 3A,D. The images made by epifluorescence microscopy of the young mycelia (Figure 3B,C) show that growing and sporulating mycelial cells of *Stachybotrys* sp. have incorporated the fluorescent DNA probes and emit strong blue or red fluorescence. The micrographs of old mycelia (Figure 3E,F) show that the fluorescent dyes were not incorporated in old, dehydrated mycelia or mature conidia. These structures emitted no strong fluorescence or liquid droplets. Figure 2 and Figure 3 illustrate that the nucleic acid stains Hoechst 3342 and propidium iodide penetrated into fresh, actively growing aerial hyphae and conidiophores of the common indoor moulds A. versicolor and *Stachybotrys* sp. The nuclei and cytoplasm in certain hyphae emitted strong blue and red fluorescence. Old, dehydrated biomasses exhibited mostly conidia and mycelia unstained by the fluorescent stains. Figure 4 shows a vital-stained inlet air filter pictured as dry (A, B) and after moisturizing (C, D, F). The dry filter showed no fungal structures, but in the moist filter the actively growing fungal structure emitted bright blue and red fluorescence.

#### 3.1.2. Formation of Guttation Droplets in Fresh and Old Plate-Grown Fungal Cultures

To facilitate monitoring the formation of guttation droplets emitted by fungal colonies of different age and different metabolic state, colonies were illuminated by UV-light. Autofluorescent droplets pictured in visible and UV-light are shown in Figure 5 and Figure 6. Figure 5 shows that blue-fluorescing exudates visible in UV-light of young cultures of *T. trixiae* LB1 (Figure 5D,E) were absent in old cultures (Figure 5A,B) and that the exudates and biomass dispersal emitted similar blue fluorescence (Figure 5C,F). Similar results were obtained for strains *T. trixiae* NJ14 and NJ22 and *T. atroviride* H1/226, 8/AM, 14/AM, Tri/335, T1/SKK, T7/SKK and KIV10. Figure 6 shows that the young cultures of the *A. versicolor*-like strain Asp2/TT5 that are yellow-green in visible light (Figure 6A) emitted bright orange fluorescence (Figure 6B) and blue-fluorescing exudates (Figure 6C). The old culture covered with dormant conidiophores (Figure 6D) emitted no fluorescence and no exudates visible in UV-light. The biomass dispersals from old and young biomass emitted similar orange fluorescence (Figure 6B,F), but the collected guttation droplets emitted strong blue fluorescence (Figure 6C,D). The age of the culture seemed to affect fluorescence emitted by the plate-grown biomass but not the fluorescence emitted by the biomass dispersed in ethanol. The same results were obtained for the *A. versicolor* strains GAS226, SL3 and the *A. versicolor*-like strains MH25 and MH35. In contrast to the *T. atroviride* strain, the substances secreted as exudates by the *A. versicolor* strain SL3 differed in fluorescence emission from the substances in the biomass dispersals (Figure 5 and Figure 6). Excitation with UV light facilitated recognition and collection of guttation droplets.

The exudates and biomass dispersals of the 10 *T. trixiae* and *T. atroviride* strains were similarly toxic to porcine kidney (PK-15) cells and boar spermatozoa and contained the same peptaibol, as already shown [16]. The orange-fluorescing biomass dispersals from *A. versicolor* were very toxic to PK-15 cells but non-toxic to sperm cells, whereas it was also shown that the blue guttation droplets were not toxic in either assay [38,41]. This indicates that the exudate and the biomass of the five *A. versicolor*-like strains contained different substances. These results indicate that young cultures emit more autofluorescence and produce more liquid exudates or guttation droplets than old cultures covered by conidiophores containing dormant conidia.

#### 3.1.3. Toxic Response in Bioassays Against Somatic Mammalian Cell Lines (PK-15 and MNA)

##### Old Biomasses of *Stachybotrys* sp. Grown on Paperboard and Fresh Biomass Grown on Malt Extract Agar (MEA)

The paperboard and the MEA supported the growth of *Stachybotrys* sp.-like colonies when inspected by microscopy (Figure 7A–F). The ten MEA plates inoculated with scrapings from the paperboard were covered by *Penicillium* sp. but all ten plates also contained 1–3 *Stachybotrys*-like colonies (Figure 7D). Three colonies were pure cultured and identified by microscopy as *Stachybotrys* sp. One strain was stored in our culture collection as HJ5 and will be identified by molecular methods (Figure 7E,F). Biomasses from the paper board and the pure cultures incubated at room temperature for 3 weeks were dispersed in ethanol (60 mg mL^−1^). Dispersals from the paperboard (*n* = 3) and pure cultures (*n* = 3) were tested for toxicity with PK-15 and murine neuroblastoma (MNA) cells. The EC_50_ concentrations for the old dry biomasses in exposed PK-15 and MNA cells were >5000 µg mL^−1^ and 1200 µg mL^−1^, respectively. The EC_50_ concentrations for the fresh biomasses from the three pure cultures were 16 µg mL^−1^. Difference between the triplicates measurements were within one dilution step of two-fold dilutions.

Old and fresh fungal biomasses growing on the old paperboard and grown for 21 days on MEA, respectively, were inspected by stereomicroscopy. Figure 7A–F shows the difference between old, dry biomass of *Stachybotrys* sp. on paperboard and actively growing, fresh biomass on MEA, respectively. The picture shows that the old, dry biomass exhibited no guttation droplets. The fresh biomass contained hyphae, conidiophores, conidia and large film surrounded liquid droplets. Microscopic evaluation of biomasses from the fresh, actively growing *Stachybotrys* sp. revealed that the volumes of guttation droplets filled with liquid exudates exceeded the volumes of the solid particles (Figure 7F).

##### Toxicity of Old Biomasses of *Stachybotrys* sp. Grown on Paperboard and Fresh Biomass Grown on Malt Extract Agar (MEA)

Ethanol cell washes of old and fresh fungal biomass collected from MEA were tested in vitro for toxicity as inhibition of proliferation with the PK-15 and MNA cell lines. The biomass dispersals contained 60 mg of fungal biomass per mL. Reduced toxicities, i.e., higher EC_50_ values in the old biomasses compared to fresh ones representing three genera of common indoor moulds: *Trichoderma*, *Chaetomium* and *Aspergillus* were recorded (Table 2). The toxicity endpoints of old biomass dispersals were >3 to 10 times higher than for the fresh biomass dispersals (Table 2). When calculating the density of conidia per microscopic fields in dispersals of 60 mg (wet weight) mL^−1^, the old, less toxic dispersals were found to contain 6 to 10 times more conidia than the more toxic, fresh dispersals. Thus, 100 times more toxicity was associated with the fresh conidia than with the old ones (Figure 8). Fresh fungal biomasses were more toxic in vitro than old dry biomasses when calculated per µg biomass (wet weight) and per conidial particle.

### 3.2. Substances Influencing Surface Tension of Water

#### 3.2.1. Actively Growing Fungal Cultures Secrete Substances Decreasing Surface Tension of Water

The investigations of fungal metabolites affecting the surface tension of liquid droplets are illustrated in Figure 9 and Figure 10. Strains toxic in the bioassays, *T. atroviride*, *Stachybotrys* sp. and a nontoxic *Rhizopus* sp. were tested. The surfactant activities and the decrease of surface tension caused by fungal metabolites are demonstrated in Figure 9. Small, regular globous liquid droplets condensed on the inner surface of the plastic lid above an uninoculated MEA plate are shown in Figure 9A,C. Liquid droplets above a fungal culture actively growing on MEA plate (<2 week old) were exposed to fungal metabolites (Figure 9B,D) and became flat and irregular in shape.

These exposed droplets also had enlarged surface areas compared to the small globous control droplets (Figure 9A,C). Droplets containing conidia of *T. atroviride* and *Stachybotrys* sp. also exhibited decreased surface tension, as indicated by the enlarged surface area (Figure 9E,F) and irregular shape of the droplet. Figure 9G,H demonstrates the surface tension decrease of a tap water droplet after addition of exudate collected from an actively growing *Rhizopus* sp. culture. These results demonstrated in Figure 8 indicated that liquid exudates and conidia emitted by actively growing fungal cultures, toxigenic as well as non-toxigenic, decreased the surface tension of condensed liquid droplets and tap water droplets.

Production of guttation droplets and dissemination of conidia are shown in Figure 10. *T. atroviride* colonies growing on gypsum liner in a sealed Petri plate emitted guttation droplets containing conidia (Figure 10A–D). The droplets captured on the inner surface of the lid are shown in Figure 10E–G. The growing hyphae extruding from the fungal droplet reduced the surface tension of the liquid droplets condensed on the lid (Figure 10F,G). The conidia germinated and produced new guttation droplets producing colonies (Figure 10H). These results indicated that guttation droplets may serve as dissemination vehicles for conidia. The results also indicate that guttation droplets contained surfactants enhancing availability of water for the germinating conidia.

#### 3.2.2. Fungal Cultures Secrete Hydrophobic Water-Repelling Substances

Droplets surrounded by hydrophobic water-repelling substances emitted by cultures of *P. expansum, T. atroviride* and *C. globosum* are illustrated in Figure 11. The panels in the upper row (Figure 11A and Figure 10B) show that the exudate droplet of *P. expansum*, colourless in visible light, contained fluorescent components when excited by UV-light. After evaporation of the droplet, dry substances were captured on the inner surface of the plastic lid. Panels in the upper row (Figure 10D–F) show that liquid droplets containing hydrophobic substances may be secreted by cultures of *Trichoderma* and *Chaetomium*. A dust mite seemed to feed on the *Chaetomium* exudate.

## 4. Discussion

Fungal pollution of indoor environment has been monitored by measuring fungal propagules in air, dust and building materials [50,51,52,53]. Monitored indicators have been the cultivable fungal conidia, evaluation of particles (conidia and hyphal fragments), presence of fungal DNA, or biomarkers as ergosterol, β-glucan, and specific mycotoxins [50,51,53,54,55,56,57]. Cell toxicity assays were used to measure intrinsic toxicity in air and dust [50,57,58]. These methods used for monitoring microbial contamination of the indoor environment have drawbacks: Measuring cultivable conidia underestimates the fungal load, while DNA-based and microscopic methods as well as the measurement of fungal biomarkers do not distinguish the live from the dead [50,52,53,54,55]. Chemical toxin analyses discriminate between anthropogenic and microbially produced toxins, but do not detect unknown toxins and do not measure bioreactivity [59,60]. Cell toxicity assays based on mammalian cells, human cells or multicellular organisms may recognize the cellular target and measure the bio reactivity of toxic substances, but do not distinguish microbial toxins from anthropogenic toxic chemicals as cleaning chemical and biocides [58,61,62,63,64,65]. None of these methods consider the metabolic activity of the indoor moulds and may also not measure relevant indicators specific for hazardous mould exposures. No guidelines for unhealthy levels of microbial contaminants in indoor air have been defined [32,33,34].

This study aimed to evaluate methods for monitoring metabolic activity in common indoor moulds (*Aspergillus, Chaetomium, Stachybotrys*, and *Trichoderma*) grown on culture media and materials used in buildings. Novel findings revealed that old, dry dormant fungal biomass emitted less fluorescence by viability staining and after UV-excitation (Figure 1, Figure 2 and Figure 3), exhibited less guttation droplets (Figure 4, Figure 5 and Figure 6) and promoted a weaker response in bioassays than actively growing fungal biomass (Table 2, Figure 8). Our results also demonstrated that surface-active agents, released from fresh biomass, migrated through the air (Figure 9, Figure 10 and Figure 11). Our results, even though based on a low number of experiments, but supported by previous studies [36,37,38], tempted us to hypothesize that actively growing fungi may have a greater influence on indoor air quality than old, dry fungal growth, irrespective of the amount of liberated conidia (Figure 8). Monitoring metabolic activity of indoor fungi may render useful information for renovation strategies of mould-infested buildings.

In bioassays using mammalian test cells, microbial secondary metabolites produced by the genera *Trichoderma, Aspergillus, Chaetomium*, and *Stachybotrys* provoked dose-dependent responses measurable as decreased sperm motility, depleted energy production, and disturbed cell proliferation [45,46,61]. Each bio reactive metabolite exhibits an in vitro toxicity profile based on difference in sensitivity, i.e., a pattern of characteristic EC_50_ concentrations obtained by the different in vitro toxicity assays. The characteristic toxicity profile indicated the biological target of the toxin as: intactness of the plasma membrane, ion fluxes and ion homeostasis [40,41,46], mitochondrial functions [40,61,66,67], transcription and translation [38,57]. The sperm motility inhibiting assay is sensitive to toxins affecting mitochondria and plasma membrane integrity, whereas the cell proliferation inhibition assays are sensitive to toxins affecting transcription and translation [40,41]. The toxins produced by the four indoor genera above provoked all toxic responses in the bioassay measuring inhibition of cell proliferation in PK-15 cells.

The low toxicity recorded for the old dry *Stachybotrys* sp. biomass scraped from the old paperboard and for the old plate-grown cultures of the indoor moulds, *T. atroviride, A. versicolor* and *C. globosum* (Figure 7, Table 2) may have three different explanations: (1) low solubility of the toxins from dry biomass in ethanol, (2) decreased bioreactivity caused by the degradation of the toxins, (3) decreased toxin contents due to emission and migration. The first conclusion has to be tested by moisturizing the old, dehydrated biomass before dispersal in ethanol. The last two explanations may indicate that finding of dormant old, dehydrated toxigenic indoor mould in a building is not necessarily connected to extreme toxicity. This information may be important for renovation work. However, it is to be kept in mind that only in vivo experiments exposing experimental animals, to known respired doses, may reveal the health effects following inhalation exposure to substances emitted by mouldy materials [68]. In vitro assays may not be acceptable as regulatory models.

Autofluorescence after UV excitation was more strongly emitted by young, fresh, actively growing fungal hyphae and could possibly be used for estimating metabolic activity on fungal hyphae growing inside buildings. UV excitation may also facilitate visualization of guttation (Figure 4 and Figure 5). Fungal species of the genera *Stachybotrys, Trichoderma, Penicillium*, and *Chaetomium* emitted the same mycotoxins in biomass dispersals and liquid droplets. In the genus *Aspergillus*, guttation droplets differed from biomass dispersals in fluorescence emission and were non-toxic in bioassays where the biomass dispersals gave toxic response (Figure 5 and Figure 6) [37,38,41]. Recent studies suggest that some mycotoxins may be synthesized from precursors in extracellular trafficking droplets, that house secondary metabolite synthesis [69,70]. This explains why substances exhibiting different toxicity and auto fluorescence may be found in liquid droplets and in conidia and hyphae. These results also indicate that auto fluorescence may facilitate visualization of aerial fungal hyphae actively growing on indoor building materials, but not that of old, dehydrated ones.

The results shown in Figure 9 and Figure 10 indicate that fungal cultures may release surfactants as well as hydrophobic water-repelling substances. Figure 11 indicates that fungal cultures emit hydrophobic substances surrounding liquid droplets as water-repelling film. These substances, surfactants and hydrophobins, may contribute to microbial indoor air pollution and influence water availability and growth conditions and germination of fungi growing on building materials [9]. Reduction of the surface tension of water may be caused by anthropogenic chemicals and/or be microbially produced, but surface-active substances exist in much higher airborne concentrations in indoor environments compared to outdoor air [24,25]. Tensiometric measurements and assays measuring plasma membrane damage [48,71] could be used for monitoring surfactant activity in indoor samples, but only chemical analyses could reveal their microbiological or anthropogenic origin [59].

Aerial hyphae of fresh, actively growing biomass stained with the viability stain in the upper rows of Figure 2, Figure 3 and Figure 4 emitted strong blue (live) and red (dead) fluorescence. However, the strong blue and red fluorescence emitted by the conidiophores, stripes and phialides in Figure 2 and Figure 3 may be caused by something else than intracellular nucleic acids. The red and blue fluorescence may not even separate dead and live fungal structures.

Dehydrated, inactive fungal structures containing dormant conidia emitting week or no fluorescence were apparently impermeable both to the live and to the dead stain (Figure 2, Figure 3 and Figure 4, lower rows) and were probably unable to emit liquid metabolites. Fungal hyphae actively growing on inlet air filters, as those shown in Figure 4D,E may be able to promote emissions of secreted liquid microbial pollutants into indoor air. Interestingly the cytoplasm and some nuclei in fresh hyphae in Figure 2, Figure 3 and Figure 4 stained red, indicating possible permeability to PI. The lipophilic DNA stain Hoechst 33342 is known to emit blue fluorescence in live and dead cells. PI is believed to cross only plasma membranes with disturbed permeability, staining DNA and RNA in affected cells red. Since fungal hyphae stained with Hoechst 33342 only emitted exclusively blue fluorescence, the red emission was concluded to be caused by intracellular PI (Figure 2, Figure 3 and Figure 4). Permeability to PI of fungal membranes may be affected by the electrical membrane potential (ΔΨ). A boosted ΔΨ might facilitate a breakthrough of propidium molecules [43]). The plasma membrane of fungal hyphae may respond to optical stimulation by raising its ΔΨ and keeping it raised till the light is switched off [1]. Apparently, the viability stain Hoechst 33342 plus PI, known to discriminate between live and dead mammalian cells [40,41,44] stained fungal hyphae according to different criteria.

## 5. Conclusions

In conclusion, our results suggest that the metabolic state and guttation influence the content of secondary metabolites (toxins, proteins and surfactants) in indoor fungal biomass and may influence their emission into indoor air. Surface-active substances, both anthropogenic and microbially produced, may influence the migration of microbial, airborne pollutants in indoor environments and enhance their toxic and allergic effects. Measuring surfactant activity in air and dust combined with toxicity assays and chemical identification of the detected surfactants and toxins may provide information useful when indoor air quality is monitored.

## Figures and Tables

**Figure 1 microorganisms-08-01940-f001:**
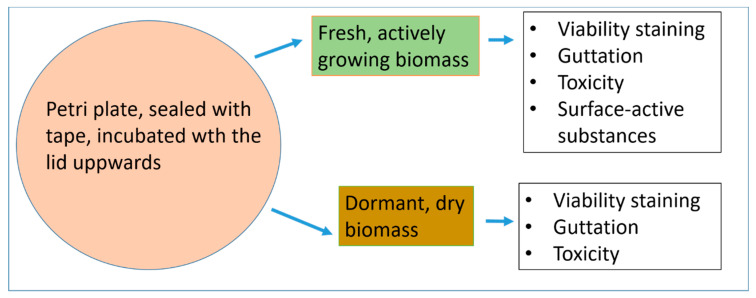
Experimental procedure for comparison of metabolic activity in fresh, actively growing and old, dormant, dry fungal biomass. The investigated biomass was produced by cultivation on malt extract agar, on gypsum liner, or on a piece of exhaust filter. Fresh, actively growing, and dormant, dry biomasses were incubated for 1 to 3 weeks, and 5 to 6 months, respectively.

**Figure 2 microorganisms-08-01940-f002:**
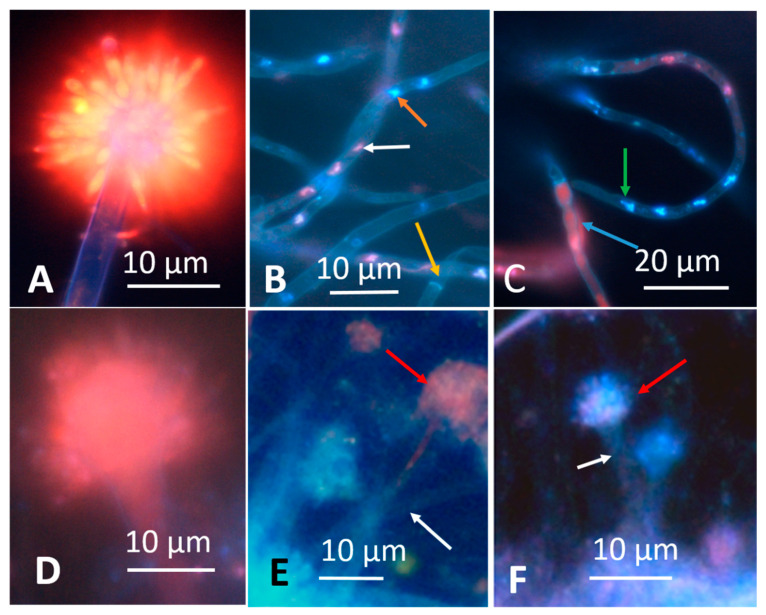
Micrographs showing young and old conidiophores and hyphae of *Aspergillus versicolor* SL/3 grown on malt extract agar and stained with the nucleic acid stain Hoechst 33342 + propidium iodide. Hoechst 33342 crosses intact and damaged plasma membranes, staining the DNA in nuclei and mitochondria in both live and dead cells blue. Propidium iodide crosses only plasma membranes with lost or exceptional integrity, and stains intracellular DNA and RNA red. The upper row (**A**–**C**) is pictured from a young (1 week), actively growing culture, whereas the lower row (**D**–**F**) shows an old (8 weeks), dehydrated, inactive culture. (**A**) conidiophore emitting bright red-yellow fluorescence in the phialides and metulae, the stipe emitting violet fluorescence; (**B**) hyphae containing blue and red fluorescing nuclei (orange and white arrows) and a septum (yellow arrow); (**C**) hyphae containing irregular dividing nuclei (green arrow) and a hyphae with red fluorescence emitting cytoplasm (blue arrow); (**D**) conidiophore emitting uniform weak fluorescence in the phialides, metulae and stipe; (**E**) non-fluorescent hyphae (white arrow) and a weakly red-fluorescing conidiophore (red arrow); (**F**) nonfluorescent stipe (white arrow) and a violet fluorescing conidiophore (red arrow).

**Figure 3 microorganisms-08-01940-f003:**
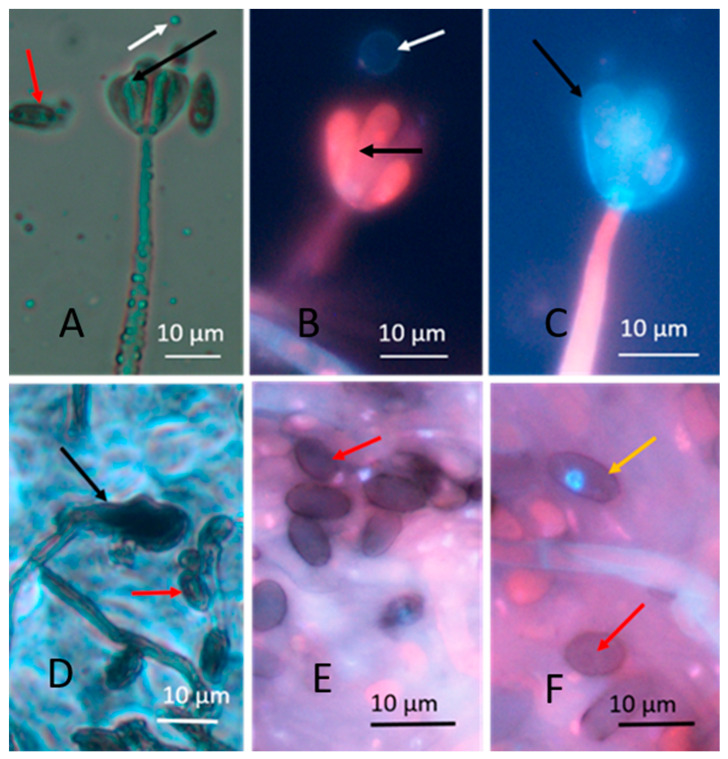
Micrographs made by phase contrast (**A**,**D**) and epifluorescence microscopy (**B**,**C**,**E**,**F**, stained with Hoechst 33342 and propidium iodide). Micrographs show young fresh (upper row) and old dormant (lower row) hyphae and conidia of *Stachybotrys* sp. strain HJ5 grown on malt extract agar. The fresh mycelium in the upper row has grown for 10 days, while the dry mycelium in the lower row for 6 months. Black and red arrows show conidiophores (**A**) and conidia (**B**), respectively. In Panels (**B**,**C**) and (**E**,**F**) the mycelia are (**A**) possible emission of liquid droplets (white arrow); (**B**) conidiophore and phialide stained red (black arrow) and an emitted particle slightly stained blue (white arrow); (**C**) blue stained phialide (black arrow); (**E**,**F**) no fluorescent mycelium, conidiophores or conidia, the dark dormant conidia (red arrows) and the dehydrated mycelia are impermeable to the fluorescent dyes, except for one conidium with a blue-stained nucleus ((**F**), yellow arrow).

**Figure 4 microorganisms-08-01940-f004:**
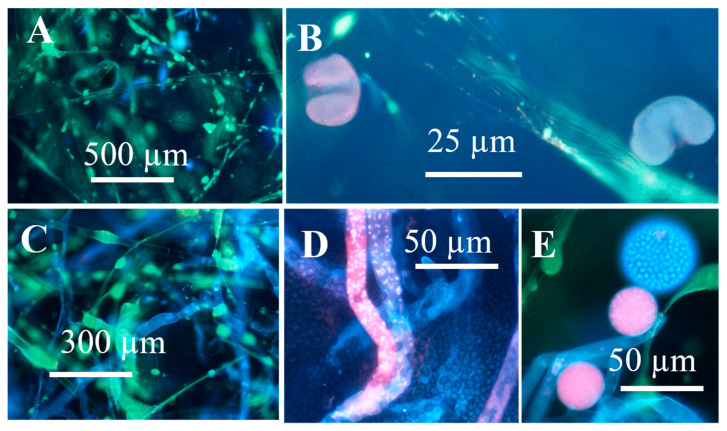
Micrographs of an inlet air filter stained with the viability stain Hoechst 33342 + propidium iodide. The upper row shows the dry filter exhibiting glass fibre and pollen grains (**A**,**B**). The lower row shows the filter after moisturizing for 48 h. The pictures show actively growing fungal structures (**C**–**E**) and dormant conidia (**D**).

**Figure 5 microorganisms-08-01940-f005:**
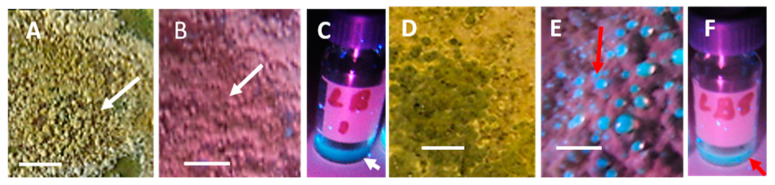
Plate-grown biomass, biomass dispersals and exudates from *Trichoderma trixiae* strain LB1 grown on malt extract agar. The panels were pictured in visible (**A**,**D**) and UV (**B**,**C**,**E**,**F**) light. (**A**,**B**) biomass of a culture in the late stationary phase (>4 weeks); (**C**) biomass lysate of a culture in the late stationary phase (>4 weeks); (**D**,**E**) biomass in the early stationary phase (1 week); (**F**) guttation droplets collected in an ampule from the plate-grown biomass in Panel B. The biomass consisted of hyphae, green conidia and blue-fluorescing guttation droplets containing liquid exudates. The bars are 10 mm.

**Figure 6 microorganisms-08-01940-f006:**
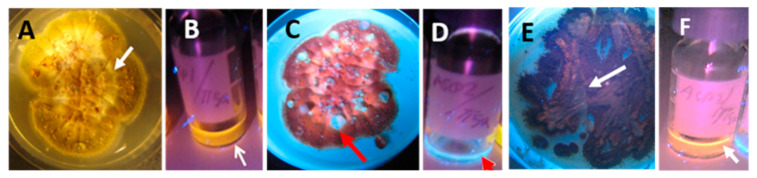
Colonies and biomass dispersals of *Aspergillus versicolor* strain Asp2/TT5a cultivated on malt extract agar, pictured in visible (**A**) and UV light (**B**–**D**,**F**). The colony cultivated for 7 days is yellow-green in visible light (**A**) and orange in UV light (**C**). Biomass scraped from the plate ((**A**), white arrow), suspended in 200 µL ethanol emitted orange fluorescence in UV light ((**B**), white arrow). The blue-fluorescing liquid droplets ((**C**), red arrow) from the colony pictured in UV-light (**C**) were collected into an ampule ((**D**), red arrow). Non-fluorescing biomass of the same strain covered by dormant conidia after 3 months of incubation ((**E**), white arrow) emitted orange fluorescence when dispersed in ethanol ((**F**), white arrow).

**Figure 7 microorganisms-08-01940-f007:**
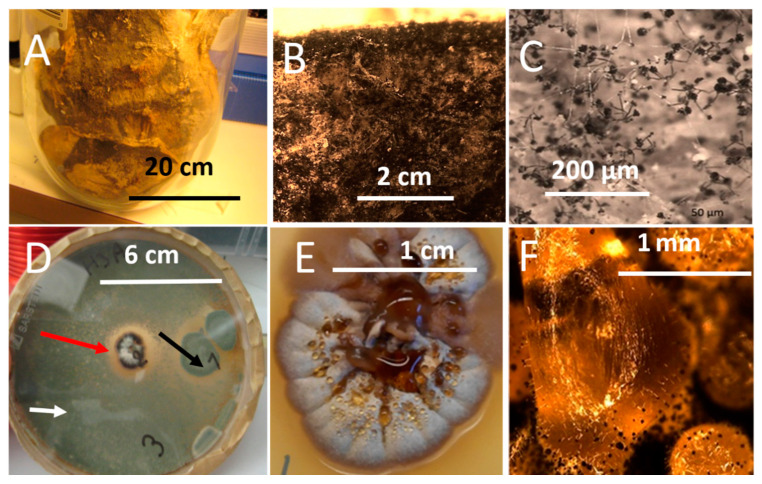
*Stachybotrys* sp. biomass on water-damaged paperboard (upper row) and on malt extract agar (lower row). The paper board was stored for <20 years in a sealed glass jar (**A**). Stereomicrographs of the paperboard showed black microbial growth (**B**) and conidiophores characteristic for *Stachybotrys* (**C**), with no exudates or vesicles visible. Panel (**D**) shows a malt extract plate inoculated with biomass scrapings from the paper board. After 3 week the plate was covered by green *Penicillium*-like growth (white and black arrows), and a black antagonistic *Stachybotrys*-like colony (red arrow) appeared. Panels (**E**,**F**) show stereomicrographs of the *Stachybotrys*-like colony, pure cultured, named HJ5 and pictured after 1 and 3 weeks of incubation, respectively. Panel (**F**) shows large liquid-containing film-surrounded droplets covered by black conidiophores.

**Figure 8 microorganisms-08-01940-f008:**
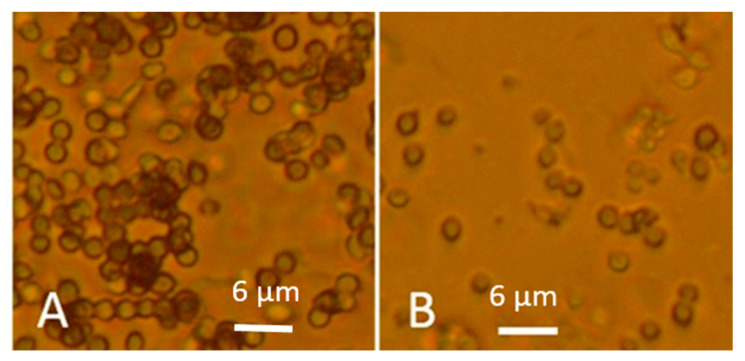
Phase contrast micrographs showing density of conidia in microscopic fields in dispersals of plate-grown biomasses (60 mg wet weight per mL) of *Aspergillus versicolor* SL/3. The dispersal of >1-year-old biomass in Panel (**A**) visualizes 120 conidia. Panel (**B**) shows the dispersal from a 2-weeks-old biomass containing 20 conidia. The panels (**A**,**B**) are representatives for 6 microscopic fields, respectively.

**Figure 9 microorganisms-08-01940-f009:**
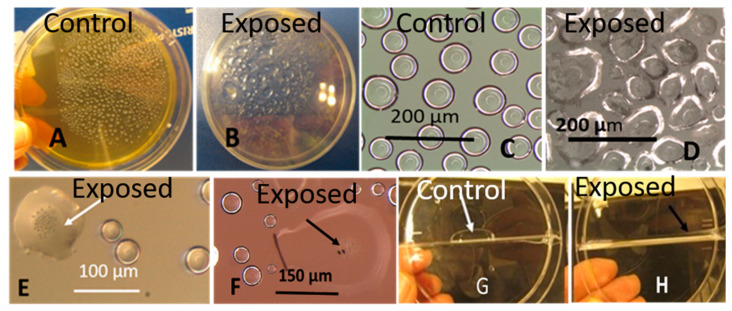
Surfactant activity of fungal exudates and conidia viewed by naked eye (**A**,**B**,**G**,**H**) and stereomicroscope (**C**–**F**). The figure shows liquid droplets condensed on the inner surface of the plastic Petri plate lid above fungal cultures incubated for 2 weeks on malt extract agar (**A**–**F**). Panels (**G**,**H**) show tap water droplets of 500 µL on a ventilation cam of a two-compartment Petri dish. The controls exhibit the liquid droplets with an undisturbed surface tension above a non-inoculated agar plate (**A**,**C**) and on the ventilation cam (**G**). Flat, enlarged, irregular droplets exposed for 2 weeks to fungal metabolites above a plate with a *T. atroviride* culture (strain T1/SKK) are shown in Panels (**B**,**D**). Flat, enlarged droplets containing conidia emitted by T1/SKK (white arrow) and of *Stachybotrys* sp. strain HJ5 (black arrow) are shown in Panels (**E**,**F**). Panel (**H**) shows the surface enlargement (black arrow) of the control droplet in Panel (**G**) (white arrow) after addition of 50 µL exudate from *Rhizopus* sp. strain MKA1.

**Figure 10 microorganisms-08-01940-f010:**
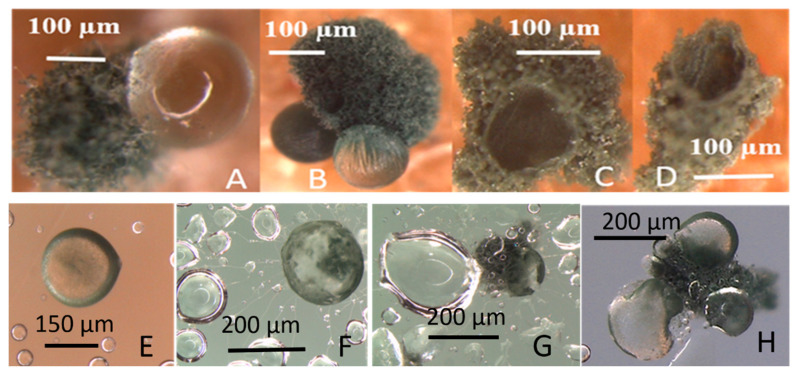
Stereomicrographs of *Trichoderma atroviride* 14/AM colonies grown for 3 weeks on a gypsum liner in a sealed Petri plate with the lid upwards. The upper row shows colonies with large guttation droplets (**A**,**B**) and dry empty membrane structures (**C**,**D**) growing on the gypsum liner. The lower row shows guttation droplets containing conidia on the inner surface of the lid. Panel (**E**) shows a fresh droplet, Panels (**F**,**G**) show droplets where conidia germinated, and hyphae extrude from the droplet. Panel (**H**) shows a new colony producing guttation droplets on the inner surface of the lid.

**Figure 11 microorganisms-08-01940-f011:**
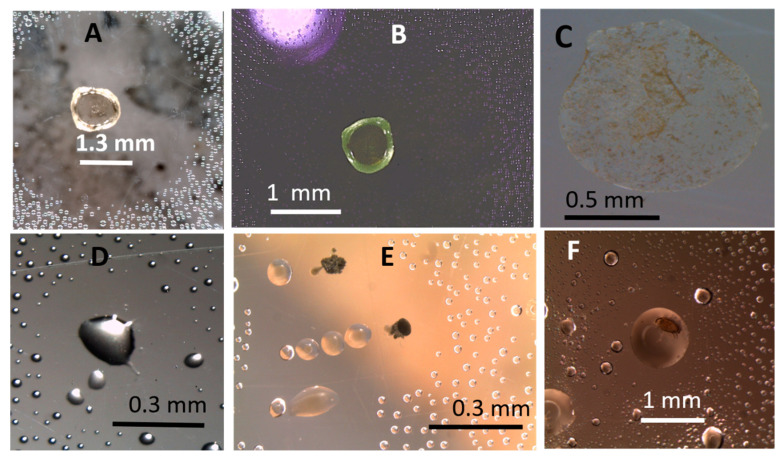
Stereomicroscopic views of water-repelling liquid droplets captured on the inner surface of plastic Petri-plate lids. The lids were covering fungal cultures incubated on malt extract agar for 4 weeks. The upper row shows droplets emitted by *Penicillium expansum* strain Rcp61. (**A**,**B**) show colourless and green fluorescence-emitting exudate droplets when pictured in visible and UV-light, respectively; (**C**) dry substances of a liquid droplet evaporated on the inner surface of the lid; (**D**,**E**) show water-repelling nature of the hydrophobic droplets emitted by cultures of *Trichoderma atroviride* 14/AM after 2 and 6 weeks of incubation, respectively; (**F**) hydrophobic droplet emitted by a 3-months-old *Chaetomium globosum* MH5 culture containing a dust mite.

**Table 1 microorganisms-08-01940-t001:** Identification and origin of the indoor fungal strains used in this study. The biomass lysate was considered toxic when 2.5 vol % decreased boar sperm motility in the BSMI assay or 5 vol % decreased proliferation of PK-15 cells in the ICP assay by >50% compared to the sham-exposed control.

Strain Code	Origin	Sample Site	Identified *	Biomass Lysates			Reference
				Toxicity		Fluorescence	
				BSMI	ICP		
*Aspergillus* section *Versicolores* unable to grow at 37 °C							
Asp2/TH	Settled dust	School, Vantaa	Morphotype	−	+	Orange	This study
GAS226	Settled dust	Office, Helsinki	DSMZ	−	+	Orange	[38,41,45]
MH25	Settled dust	University, Espoo	Morphotype	−	+	Orange	[41]
MH35	Settled dust	University, Espoo	Morphotype	−	+	Orange	[41]
SL/3	Fall out plate	Apartment, Helsinki	DSMZ	−	+	Orange	[37,38,41,45]
*Chaetomium globosum* unable to grow at 37 °C							
MH5	Settled dust	University, Espoo	*tef1α* (MT498108)	+	+	Blue-green	[45]
MO9	Settled dust	Piggery, Orimattila	*tef1α* (MT498106)	+	+	Blue-green	[45]
2b/26	Settled dust	Apartment, Vantaa	*tef1α* (MT498110)	+	+	Blue-green	[45]
2c/26	Settle d dust	Apartment, Vantaa	*tef1α* (MW310244)	+	+	Blue-green	[45]
*Trichoderma atroviride* unable to grow at 37 °C							
H1/226	Fallout plate	Office, Helsinki	*tef1α* (MH176994) ITS (KM853017)	+	(+)	Blue	[37]
T1/SKK	Exhaust filter	School, Vantaa	ITS (MW306736)	+	(+)	Blue	[42,49]
T7/SKK	Exhaust filter	School, Vantaa	ITS (MW306737)	+	(+)	Blue	[42,49]
8/AM	Exhaust filter	University, Espoo	*tef1α* (MH176996) ITS (MH158553)	+	(+)	Blue	[37]
14/AM	Exhaust filter	University, Espoo	*tef1α* (MH176997) ITS (MH158554)	+	(+)	Blue	[37]
Tri/335	Settled dust	University, Espoo	*tef1α* (MH176998)	+	(+)	Blue	[37]
KIV10	Andersen impactor	School, Lahti	*tef1α* (MH176999)	+	(+)	Blue	[37]
*Trichoderma trixiae* unable to grow at 37 °C							
LB1	Settled dust	Apartment, Helsinki	*tef1α* (MH177001) ITS (MH158556)	+	(+)	Blue	[37]
NJ14	Settled dust	Ice Rink, Nivala	*tef1α* (MH177002) ITS (MH158557)	+	(+)	Blue	[37]
NJ22	Settled dust	Ice Rink, Nivala	*tef1α* (MH177003) ITS (MH158558)	+	(+)	Blue	[37]
*Penicillium expansum* unable to grow at 37 °C							
Rcp61	Cork liner	University, Espoo	ITS (MK201596), WI	(+)	+	Blue	[38]
*Rhizopus* sp. grew at 37 °C							
MKA1	Exhaust filter	School, Vantaa	Morphotype	−	−	Blue	This study
*Stachybotrys* sp. unable to grow at 37 °C							
HJ5	Paperboard	Apartment, Mäntsälä	Morphotype	(+)	+	Blue	This study

* numbers in parentheses are GenBank accession numbers of sequences, + test with most toxic response, EC_50_ for + < (+) < −, WI: Westerdijk Institute.

**Table 2 microorganisms-08-01940-t002:** Toxic responses in in vitro assays with somatic mammalian cell lines PK-15 and MNA of actively growing, fresh biomass and dry, old fungal biomass grown on malt extract agar.

	EC_50_ µg Fungal Biomass (Wet Weight) mL^−1^
Age of Biomass	2 Weeks	1 Year
	PK-15	MNA	PK-15	MNA
*Trichoderma atroviride*		
T1/SKK	100	100	2500	2500
T7/SKK	130	130	1500	1500
H1/226	100	100	5000	5000
*Aspergillus versicolor*		
SL/3	100		>320	
*Chaetomium globosum*			
9/MO	30		600	
2b/26	40		2500	
2c/26	60		400	
MH5	50		600

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
