# Peer review of "Bioreactivity, Guttation and Agents Influencing Surface Tension of Water Emitted by Actively Growing Indoor Mould Isolates"

_microorganisms, 2020, doi:10.3390/microorganisms8121940_

Round 1

Reviewer 1 Report

The manuscript  entitled “Bioreactivity, guttation and agents influencing surface tension of water emitted by actively growing indoor mould isolates” have demonstrated develop and evaluate fast microscopic and toxicological methods for monitoring metabolic activity in fungal colonies, and to reveal the metabolic differences between fresh, actively growing and old colonies. The study was written carefully and well in terms of language.

Authors should correct manuscript according to the suggestion and completed some information.

Minor issues:

Materials and methods:

Line 91 - 92: in my opinion, sequences and primers ITS and  tef1α should be given,

what database were the obtained sequences compared to? please provide the % similarity of the isolated strain with other strains of  from database and their numbers.

Line 118 – 121: preincubation parameters for BSMI and PK-15  cell cultures (e.g medium, cells density, time of incubation) should be given

Line 284 -286: based on resazurine reduction factors authors should provide addition information e.g. cells viability (in %)

Author Response

Requests from reviewer I, 1-4:

  • Line 91 - 92: in my opinion, sequences and primers ITS and tef1α should be given,

Answere: The primers and databases are given in line 91-96.

  • What database were the obtained sequences compared to? please provide the % similarity of the isolated strain with other strains of from database and their numbers.

 Answere: The accession numbers of the sequences are inserted in Table 1. The sequences were compared by BLAST with the sequences in the NCBI database. The BLAST  comparison can be done based on the accession numbers given in Table 1.

  • Line 118 – 121: preincubation parameters for BSMI and PK-15 cell cultures (e.g medium, cells density, time of incubation) should be given.

Answere: The semen extender and cell culture medium,  density of exposed sperm cells and somatic cells, as well as the incubation times in te bio assays are inserted in lines 125-130 (highlighted yellow).

  • Line 284 -286: based on resazurine reduction factors authors should provide addition information e.g. cells viability (in %)

Answere: The % corresponding to cell viability is given in lines 141-146 (highlighted yellow)

Reviewer 2 Report

The aim of our research was to create rapid microscopic and toxicological methods for monitoring metabolic activity in fungal colonies and to reveal metabolic differences between fresh, actively growing and old, dried, dormant colonies growing on building materials and culture media. 

The purpose of the work is clearly stated. The conclusions of the conducted research are clear and result from the obtained research results. The material used for the research is sufficient, the research methods have been selected appropriately. The arrangement of the figures is clear and presents the obtained results very well.
Discussing the results against the background of other authors is very detailed. The publications cited by the authors of the article are well selected. For the most part, the authors refer to the latest knowledge published in renowned scientific journals.

Author Response

we found no request from reviewr II